# Ternary Cu(II) Complex with GHK Peptide and *Cis*-Urocanic Acid as a Potential Physiologically Functional Copper Chelate

**DOI:** 10.3390/ijms21176190

**Published:** 2020-08-27

**Authors:** Karolina Bossak-Ahmad, Marta D. Wiśniewska, Wojciech Bal, Simon C. Drew, Tomasz Frączyk

**Affiliations:** 1Institute of Biochemistry and Biophysics, Polish Academy of Sciences, Pawińskiego 5a, 02-106 Warsaw, Poland; karolina.bossak@gmail.com (K.B.-A.); marta.d.wisniewska@gmail.com (M.D.W.); wojciech.bal.ibb@gmail.com (W.B.); scdrew1@gmail.com (S.C.D.); 2Department of Medicine (Royal Melbourne Hospital), The University of Melbourne, Melbourne 3010, Australia

**Keywords:** copper, ternary complex, imidazole ligands

## Abstract

The tripeptide NH_2_–Gly–His–Lys–COOH (GHK), *cis*-urocanic acid (*cis*-UCA) and Cu(II) ions are physiological constituents of the human body and they co-occur (e.g., in the skin and the plasma). While GHK is known as Cu(II)-binding molecule, we found that urocanic acid also coordinates Cu(II) ions. Furthermore, both ligands create ternary Cu(II) complex being probably physiologically functional species. Regarding the natural concentrations of the studied molecules in some human tissues, together with the affinities reported here, we conclude that the ternary complex [GHK][Cu(II)][*cis*-urocanic acid] may be partly responsible for biological effects of GHK and urocanic acid described in the literature.

## 1. Introduction

The peptide Gly–His–Lys (GHK) is a native constituent of human blood [1]. It has numerous actions, including wound-healing [2], anti-inflammatory [3], and anti-anxiety [4] activities which may result from the interaction with unidentified receptors. Importantly, GHK modulates the expression of many genes [5]. It was proposed that almost all of the observed effects of the peptide are evoked by a GHK–copper(II) complex [6].

Urocanic acid (UCA) is a component of natural moisturizing factor (NMF) in the uppermost layer of the skin (stratum corneum) [7]. It is also found in blood [8] and neurons [9]. UCA is the product of histidine deamination by histidine ammonia–lyase (histidase). The formed *trans* isomer (*trans*-UCA) isomerises to *cis*-UCA upon exposure to ultraviolet radiation (Scheme 1) [10]. Both isomers accumulate in stratum corneum, reaching millimolar concentrations [11,12], and are present in plasma at micromolar concentrations [8]. Recently, it was found that UCA can cross the blood–brain barrier and be transported into neurons. Furthermore, histidase activity has also been found in neurons [9]. Importantly, there is a positive correlation between the content of histidine (the substrate for production of UCA) and wound healing [13,14]. UCA has many functions, including skin hydration maintenance, pH regulation, UV protection, and immunosuppression [10,15,16].

We have previously shown that UCA can bind Ni(II) ions [17]. Bearing in mind the similarity of Ni(II) and Cu(II) complex formation with many low molecular weight compounds, copper binding by UCA is conceivable.

Interestingly, GHK and UCA coexist in some human tissues and have common biological activities, such as influencing the immune and nervous systems. Because each can also bind metal ions, we hypothesised that biologically active forms of GHK and UCA may include a ternary complex with Cu(II) ions.

In this work, we proved that GHK forms ternary Cu(II) complexes with *cis*-/*trans*-UCA. The results obtained by UV-vis and circular dichroism (CD) spectrophotometry, room-temperature electron paramagnetic resonance (EPR), and potentiometry suggest that such complexes can be present in the human body.

## 2. Results and Discussion

### 2.1. Interaction of Cu(II) with GHK

In order to thoroughly characterise the ternary interaction of urocanic acid, GHK, and Cu(II) we decided to revisit the Cu(GHK) coordination via spectroscopic and potentiometric studies. This step is necessary for precise comparison of experiments involving Cu/GHK and Cu/GHK/UCA. On that note, we also elaborated on Cu(GHK)_2_, the complex that was reported previously by Conato et al. [18].

We report here a refined potentiometric model that incorporates *mono*- and *bis*- Cu(II) complexes with GHK and is supported by the spectroscopic characterization of these species. GHK has four exchangeable protons, attached (in the decreasing order of p*K* values) to the lysine side chain nitrogen, N-terminal amine, histidine imidazole (Im) nitrogen, and C-terminal carboxylate. Their protonation constants reported herein (Appendix A) are in agreement with the literature data [19]. Potentiometry of Cu/GHK identified six Cu(II) species, namely Cu(II)^aq^, CuH(GHK), Cu(GHK), CuH_−1_(GHK), CuH_−2_(GHK), and CuH_2_(GHK)_2_. The calculated logβ values and the ascribed protonation events are presented in Appendix A. We validated our potentiometric model by combining two sets of pH-metric titrations with spectroscopic detection, the first set using 0.95 mM GHK in the presence of 0.8 mM CuCl_2_ (Appendix A), and the second using a 20-fold molar excess of GHK (10.63 mM GHK and 0.5 mM CuCl_2_). In an equimolar Cu/GHK solution, the variation in absorbance at 605 nm at low pH yielded a Hill coefficient of 2.03 ± 0.32, suggesting the cooperative formation of two species that can be ascribed to CuH(GHK) and Cu(GHK) stoichiometries. These 3N complexes differ only by the protonation state of the C-terminal carboxyl group and have identical *d–d* bands. Above pH 4.5, CD spectra obtained with a high excess of GHK exhibited a more prominent blue-shift compared with the equimolar complex. This suggests the coordination of an external nitrogen donor at the fourth planar Cu(II) site. Such complex corresponds to CuH_2_(GHK)_2_ stoichiometry. However, the observed changes in the electronic spectra are very similar to those seen previously in ternary complexes of other Xaa-His peptides (where Xaa is any amino acid residue except Pro) with imidazole (3+1N coordination, where three nitrogen atoms, NH_3_^+ (N-term)^, N^amide^, and N^Im^, are from the first peptide molecule, and one nitrogen atom is N^Im^ from the second peptide molecule) [20,21]. Therefore, we assign the 3+1N coordination to this complex and propose to call it “auto-ternary”, with the actual Cu(GHK)(H_2_GHK) stoichiometry. Two additional deprotonations occur in the *mono*-complex at alkaline pH. The first probably corresponds to a water molecule (p*K* = 9.61) within the equatorial plane of the Cu(GHK) species [20,21]. Another is associated with the ε-amino nitrogen (p*K* = 10.77) from a side chain of Lys residue. The aforementioned assignment of the deprotonation to a water molecule instead of an imidazole N1 from another Cu(GHK) complex (with concomitant polynuclear species formation) is rationalized by the lack of drastic changes of CD parameters that would be expected upon formation of imidazole-bridged dimeric or tetrameric species [21,22,23].

The competitivity index (CI) [24,25] is a convenient parameter to compare the apparent affinities of complexes with various stoichiometries and protonation states. The value of CI equals to logβ of a complex MZ of a metal ion (M) with a theoretical ligand Z capable of outcompeting 50% of metal ion from the tested ligand or system of ligands at given conditions, such as specific pH and concentrations of reactants, when [Z] = [L] (L is a ligand to be compared). The Z molecule is supposed to bind only in 1:1 stoichiometry and to lack any deprotonating groups. This is equivalent to the assumption that $\sum_{ijkl}\left(\left[M_{i}H_{j}L_{k}A_{l}\right]\right)=\left[MZ\right]$. In simple cases, CI corresponds to conditional stability constant, ^c^*K*. We used potentiometric data obtained for Cu/GHK and the CI method to calculate the conditional stability constants for the equimolar Cu(II)/GHK complex at [GHK] = [Cu] = 1 µM at pH 7.4, ^c^*K*_7.4_ = 4.17 × 10^12^ M^−1^ and pH 6.5, ^c^*K*_6.5_ = 4.79 × 10^10^ M^−1^, corresponding to conditional dissociation constants of 0.24 and 20.9 pM, respectively. The CI value increases for higher excess of GHK over Cu(II), resulting from the formation of the auto-ternary Cu(GHK)_2_ complex. However, the noticeable increase of CI was seen for millimolar concentrations of GHK (Appendix A).

Several groups have previously studied Cu/GHK complexes [18,26,27,28]. Our calculated protonation and formation constants values are closest to those obtained by Lau et al. [28] and Conato et al. [18], two groups that also utilised potentiometry to calculate the binding constants. Values published by others [26,27] (using EPR and ITC) are slightly higher than ours, most likely arising from the omission of Cu(GHK)_2_ stoichiometry in their models.

As mentioned above, GHK coordinates copper via a tridentate nitrogenous chelate, with the fourth binding site, being open to coordination by an external ligand. This ligand can be a solvent molecule or, in the case of Cu(GHK)_2_, an imidazole nitrogen of a second GHK molecule. To characterise the formation of this complex at physiological pH, we followed the titration of GHK into Cu(GHK) at pH 7.4 using UV-vis, EPR, and CD spectroscopies (Figure 1, Appendix A), together with isothermal titration calorimetry (ITC, Appendix A, Appendix A). Excess GHK caused a blue-shift of the absorbance maximum by 27 nm in the UV-vis spectra (Figure 1A) and by 28 nm in the CD spectra (Figure 1B). Room temperature EPR spectra revealed the presence of two motionally-averaged species delineated by clear isosbestic points (Figure 1C) confirming the existence of just two Cu(II) coordination modes. Simulations of the EPR spectra, including ligand hyperfine structure, were consistent with the assignment of 3N and 4N species for Cu(GHK) and Cu(GHK)_2_, respectively (Appendix A, Appendix A). Global fitting of all spectroscopic data yielded a value of ^c^*K*_7.4_ = 237 ± 5 M^−1^ for the Cu(GHK)_2_ complex (Figure 1D,E), while independent analysis of the ITC titrations yielded ^c^*K*_7.4_ = 265 ± 46 M^−1^ (Appendix A). This value is significantly lower from the value ca. 500 M^−1^ that could be inferred from potentiometric stability constants reported by Conato et al. [18]. The retrospective analysis of experimental conditions in theirs, as well as our study indicates that such weak interactions could not be reliably determined by potentiometry, and our spectroscopic determination is the only valid approach.

### 2.2. Interaction of Cu(II) with UCA

By analogy with the formation of Cu(GHK)_2_, other external ligands could contribute to coordination sphere of Cu(GHK). Imidazole-based ligands can increase the overall affinity of metal complexes containing tridentate ligands by several orders of magnitude [20,29,30,31,32,33]. Urocanic acid, a by-product of L-histidine, contains an imidazole ring that could form a stable ternary Cu(II) complex with GHK in a manner similar to other Xaa-His peptides such as Trp–His–Trp–Ser–Lys–Asn–Arg, Gly–His–Thr–Asp, and Ala–His–His [20,21,34].

A series of control experiments was performed for both *cis*- and *trans*-UCA in presence of Cu(II). We have previously characterised the interaction of *cis*- and *trans*-UCA with Ni(II) ions and found out that the complexes were sufficiently stable to be of potential biological relevance [17]. Of the two isomers, *cis*-UCA binds Ni(II) ions more strongly owing to the chelate effect of its imidazole nitrogen and carboxyl oxygen. Thus, we focused on the binary and ternary Cu(II) complexes of *cis*-UCA and GHK. Logarithmic formation constants for Cu/*cis*-UCA complexes are provided in Table 1, and spectroscopic data used to validate the potentiometric model are shown in Figure 2.

As expected based on the Irving–Williams series [36], the logβ value of Cu(*cis*-UCA) is higher than that of Ni(*cis*-UCA) (4.941 vs 3.406) [17]. Analogously, the formation constant of Cu(*cis*-UCA)_2_ is higher than that of Ni(*cis*-UCA)_2_ (8.76 vs. 6.239). Similar to the Ni(II) complex, the binding of the second *cis*-UCA molecule is weaker compared with the first (by 1.12 log units). This can be attributed to both a decreased number of bidentate binding modes for the second ligand and to the repulsion between the carboxyl groups of the two *cis*-UCA molecules. In the octahedral approximation of the complex structure, the first factor equals 0.38 (log(24/10)), resulting from lowering of the binding modes from 24 for the first molecule to 10 for the second one. Thus, the repulsion between carboxyl groups contributes stronger to weakening of the binding of the second *cis*-UCA molecule (0.74 log unit). The pH-dependent binding constant for the 1:1 Cu(*cis*-UCA) complex are ^c^*K*_7.4_ = 7.16 × 10^4^ M^−1^ and ^c^*K*_6.5_ = 3.16 × 10^4^ M^−1^, respectively, corresponding to conditional dissociation constants 14.0 µM and 31.6 µM, respectively. However, as concentrations of *cis*-UCA are usually higher than the concentration of exchangeable Cu(II), the Cu(*cis*-UCA)_2_ complex is promoted increasing the apparent affinity of this molecule to Cu^2+^ ions. Indeed, CI values for such complexes for 1 µM Cu(II) and millimolar *cis*-UCA levels are enhanced by at least one log unit (Appendix A). Lowering the pH slightly diminishes the strength of Cu(II) binding (by less than one log unit). The highest known concentrations of *cis*-UCA occur in the stratum corneum, and thus Cu(*cis*-UCA) complexes may occur mainly in the skin, however, a micromolar dissociation constant makes the Cu(*cis*-UCA) complex formation in blood plausible.

Cu(*trans*-UCA) was prone to precipitation under the same experimental conditions as Cu(*cis*-UCA), which prevented proper characterization of the binary complex. This precipitation may have resulted from lower stability of the Cu(*trans*-UCA) complex, leading to formation of insoluble Cu(OH)_2_ at pH > 5. Indeed, control experiments confirmed that a molar excess of *trans*-UCA promoted the formation of a more stable CuL_2_ complex.

### 2.3. Ternary Complex Formation of Cu/GHK/Imidazole

To quantify the ternary interaction of imidazole-based ligands with the Cu(GHK) complex at pH 7.4, we titrated up to 40 molar equivalents of Im into solutions containing 0.5 mM Cu(II) and 0.63 mM GHK and monitored the corresponding changes in the UV-vis, CD, and EPR spectra (Figure 3). Increasing concentrations of Im resulted in a blue-shift of the *d–d* transitions in the absorbance (35 nm), and CD (43 nm) spectra (Figure 3A,B). However, we did not observe any decrease in the overall ellipticity, indicating that a Cu(GHK)(Im) complex was formed rather than the achiral Cu(Im)_n_ complexes (which are CD-silent). The isosbestic point at 601 nm (UV-vis) and the isodichroic point at 581 nm (CD) showed that Cu(GHK)(Im) was formed by the substitution of equatorial water in Cu(GHK) with Im. Room temperature EPR spectra revealed the presence of two motionally-averaged species delineated by clear isosbestic points (Figure 3C) confirming the existence of just two Cu(II) coordination modes, as in the Cu(GHK)/GHK experiment. Simulations of the EPR spectra, including ligand hyperfine structure, were consistent with the assignment of 3N and 4N species for Cu(GHK) and Cu(GHK)(Im), respectively (Appendix A). The rotational correlation time of Cu(GHK)(Im) was comparable with that of Cu(GHK), as expected based upon their similar molecular weight. The magnetic parameters of Cu(GHK)(Im), however, were almost identical to those characterising Cu(GHK)_2_, consistent with their common first coordination sphere. Further evidence for the proposed structure of the ternary species was obtained by substituting ^15^N^Im^ with ^14^N^Im^, which produced a change in the ligand hyperfine pattern expected for an equatorial Im ligand (Appendix A). Global fitting of all spectroscopic data yielded a value of ^c^*K*_7.4_ = 725 ± 22 M^−1^ (Figure 3D,E) for Cu(GHK)(Im), which is comparable to the values previously reported for the Xaa-His peptides Trp–His–Trp–Ser–Lys–Asn–Arg (1022 ± 70 M^−1^) and Gly–His–Thr–Asp (440 ± 14 M^−1^) [20,21]. We also determined a value of ^c^*K*_7.4_ = 532 ± 44 M^−1^ for Cu(GHK)(Im) complex using ITC (Appendix A). This is lower than the value obtained using spectroscopic methods, and it may stem from using HEPES buffer in ITC experiments.

Additionally, analysis of the ternary system was performed using CD and UV-vis pH-metric titrations of 0.95 mM GHK, 0.8 mM CuCl_2_, and 50 mM Im, alongside a series of potentiometric titrations (Appendix A) in ratios of 1:0.9:4, 1:0.9:6, and 1:0.9:8 (GHK):(Cu):(Im). At pH below 5, five copper species were identified, namely an aqua ion (absorbance maximum at 816 nm), two imidazole copper complexes, Cu(Im) and Cu(Im)_2_, and two 3N peptidic complexes, CuH(GHK) and Cu(GHK). Being achiral, the imidazole complexes are silent in CD spectra but produced a minor shoulder at 730 nm in UV-vis spectra at low pH. An increase of pH leads to the onset of Cu(GHK) dominance until an external imidazole begins to displace the complex’s equatorial water ligand. This is visible by a 43 nm shift of the *d–d* band in CD spectra (Appendix A). Under the given conditions (0.95 mM GHK, 0.8 mM CuCl_2_, 50 mM Im, at pH 7.4), the ternary complex accounts for 96.5% of the available copper, while 3.4% and 0.2% are bound to Cu(GHK) and Cu(GHK)_2_, respectively. Slightly less of the ternary complex (89.5%) is formed at pH 6.5, with Cu(GHK) accounting for 10.1% of available copper, the residual 0.4% being bound by Cu(GHK)_2_ (Appendix A).

### 2.4. Ternary Complex Formation of Cu/GHK/UCA

Knowing how imidazole interacts with Cu(GHK), we proceeded with the Cu(GHK) and UCA experiments. Due to the complexity of biological interactions and the possible translocation of UCA from stratum corneum to other locations that differ in pH, it is important to characterise the species distribution of the Cu/GHK/UCA system across a wide pH range. Because of the stronger interaction of Cu(GHK) with the *cis*-UCA isomer, and the aforementioned lower stability of Cu(*trans*-UCA), we focused on Cu/GHK/*cis*-UCA interactions.

First, we used potentiometry to characterise ternary Cu(GHK)(UCA) complex formation. We then validated the binding model using UV-vis and CD pH-metric titrations (Appendix A). Seven copper species were identified, namely the free copper aqua ion, Cu(*cis*-UCA), Cu(*cis*-UCA)_2_ (minor), CuH(GHK), Cu(GHK), Cu(GHK)(*cis*-UCA) and a minor CuH_2_(GHK)_2_ species. The logβ value for the ternary Cu(GHK)(*cis*-UCA) is 19.29(2). At low pH (for 0.95 mM GHK, 0.8 mM Cu, and 6 mM *cis*-/*trans*-UCA), CD spectra of both Cu/GHK/*cis*-UCA and Cu/GHK/*trans*-UCA show *d–d* band at 606 nm, due to the prevalence of Cu(GHK) (Figure 4A,B). Simultaneously, a small proportion of binary urocanic acid complexes manifested as a reduced ellipticity compared with the Cu/GHK sample (Figure 4C,D). Both Cu(UCA) and Cu(UCA)_2_ are CD silent, but noticeable in the absorbance spectra around 700 nm (Appendix A, Appendix A, *cf*
Figure 2). With increasing pH, the Cu(GHK) species begins to dominate but is quickly replaced above pH 4.9 by Cu(GHK)(UCA). Evidence for the latter species is provided by the blue-shift of *d–d* band corresponding to the 3N coordination sphere of Cu(GHK) (Figure 4A,B), and a comparison of the ellipticity of Cu/GHK and Cu/GHK/UCA samples at 650 nm (Figure 4C,D). The Cu(GHK)(UCA) complex is formed between pH 5–10 for both *cis*- and *trans*-UCA, as deduced from the reduced ellipticity at 650 nm in comparison with binary Cu(GHK). Although this description is qualitative, it can be estimated that *trans*-UCA forms less ternary complex than *cis*-UCA under the same experimental conditions, and thus it forms lower-stability ternary complex.

In order to quantify the ternary complex formation for both *cis*- and *trans*-UCA, titrations of Cu/GHK with *cis*- and *trans*-UCA at pH 6.5 and 7.4 were performed to obtain conditional binding constants. Addition of UCA to solutions of Cu(GHK) at pH 6.5 and 7.4 created a blue-shift in both CD and UV-vis spectra (Figure 5 and Appendix A) by ca. 46 nm for *cis*-UCA and 39 nm for *trans*-UCA suggesting coordination of an additional nitrogen ligand. Because Cu(II) and GHK were almost equimolar, the formation of Cu(GHK)_2_ is limited and cannot be the cause of blue-shift. Sole competition between Cu(GHK) and Cu(UCA)/Cu(UCA)_2_ at pH 6.5 and 7.4 can be also excluded since the achiral complexes of urocanic acid are CD-silent and thus competition would cause an overall decrease in the observed ellipticity of the Cu(GHK) complex. The concentration dependence of the ellipticity and absorbance changes with respect to the titrated ligands shows that *trans*-UCA did not reach equilibrium at the same concentration as *cis*-UCA, suggesting *trans*-UCA has a lower affinity for Cu(GHK) than *cis*-UCA. Indeed, the calculated conditional stability constants from a global fit of CD and UV-vis spectra of titrations with *cis*- and *trans*-UCA (Figure 5 and Appendix A, Table 2) confirm this observation.

At pH 7.4, *cis*-UCA binds Cu(GHK) with ^c^*K*_7.4_ 540 ± 17 M^−1^, similar to imidazole, whereas the binding of *trans*-UCA is almost three times weaker with ^c^*K*_7.4_ 200 ± 10 M^−1^. This means that when both GHK and UCA are co-localised in the human body (e.g., in the blood), *cis*-UCA is favoured to form a ternary complex. Lowering the pH to 6.5 changes the situation, since the conditional binding constant for *trans*-UCA does not change (^c^*K*_6.5_ = 186 ± 10 M^−1^), but *cis*-UCA binding to Cu(GHK) is much reduced (^c^*K*_6.5_ = 345 ± 26 M^−1^). Thus, in the upper layers of the skin, where the pH is lower, formation of a ternary complex does not greatly favour *cis*-UCA over *trans*-UCA. The observed difference in pH sensitivity can be attributed to higher basicity of the imidazole moiety in *cis*-UCA than *trans*-UCA (p*K*_a_ values of 6.74 and 5.83, respectively) [17].

### 2.5. Biological Relevance

We chose pH values of 7.4 and 6.5 to represent the pH of the blood and the skin, respectively. UCA naturally occurs in the skin. GHK is probably released locally during collagen or SPARC (Secreted Protein Acidic and Rich in Cysteine) proteolysis caused by skin damage [37,38]. The pH at the surface of healthy skin is relatively low (ca. 5–6). However, there is a pH gradient (increasing up to 7.4) through all layers of the skin [39]. Importantly, the pH of the skin immediately increases after damage, being closer to 7.4 [39], and even above 8 [40,41]. Persistence of high pH is characteristic of a chronic wound [40,41]. Taking this into account, the known function of GHK as a healing factor may result from a stable ternary Cu(GHK)(*cis*-UCA) complex at pH 7–8. Binding of the ternary partner (*cis*-UCA) increases the apparent affinity of GHK for Cu(II) (Figure 6). Thus, the presence of such a complex after skin damage is plausible. The wound healing process involves several steps: haemostasis, inflammation, proliferation, and tissue remodeling [39]. The role of *cis*-UCA in immunosuppression has been revealed [10]. Therefore, it seems probable that this molecule, perhaps in the form of a ternary complex, prevents the inflammation step to develop into the chronic state. In this context, it is interesting that supplementation with histidine (a substrate for UCA synthesis) accelerates wound healing [14]. Furthermore, a deficiency in histidine was observed in skin wounds [13].

The formation of the ternary complex may also be of relevance to recent usage of UCA or Cu/GHK in cosmetics, anti-allergic or wound-healing-promoting materials, involving GHK immobilized on polymers or nanoparticles, or within liposomes [2,6,42,43,44,45,46].

Literature data regarding the concentrations of main components of natural moisturizing factor (NMF) in stratum corneum (43 mM serine, 30 mM glycine, 23 mM pyroglutamic acid, 18 mM alanine, 14 mM lactic acid, and 14 mM *cis*-UCA) [11,12] and Cu(II) complex formation constants [47,48] can be used to simulate the Cu(II) speciation. For this purpose, we calculated the molar fractions of Cu(II) species at pH 7.4 (representing the conditions found in wounds; Figure 7), and 6.5 (corresponding to healthy skin; Appendix A). We also assumed a GHK concentration of 0.6 µM, as found in human plasma. One can speculate that the local concentration of GHK may be even higher at the site of skin damage during wound healing. Furthermore, the application of wound-healing materials containing GHK as the active substance can lead to an increase in the concentration of this peptide. Thus, we also calculated the Cu(II) speciation assuming 6 and 60 µM GHK. At pH 7.4 and the lowest concentration of GHK, the majority of Cu(II) is complexed with serine and glycine. However, 46% of copper bound to GHK is in the ternary Cu(GHK)(*cis*-UCA) complex. The latter percentage does not change for higher concentrations of GHK, but Cu(GHK)(*cis*-UCA), Cu(GHK)(carboxylates), (including carboxylic acids and amino acids binding monodentately via their carboxylic functions) [20], and Cu(GHK) start to dominate the overall Cu(II) speciation (Figure 7). Only slightly lower concentrations of Cu(GHK)(*cis*-UCA) and Cu(GHK) are predicted at pH 6.5 (Appendix A).

GHK and UCA also are present in the blood, although at lower concentrations [8]. UCA crosses the blood–brain barrier and was even found in cerebrospinal fluid and neurons [9]. GHK was also found to be transported into the brain [6]. It remains of great interest to determine the relevance of the ternary Cu(GHK)(*cis*-UCA) complex to the neuronal environment.

Another important aspect of our findings is the ternary complex formation of Cu(GHK) with imidazole donors in general. The imidazole ring of the histidine residue in proteins and peptides can also form a ternary complex with an apparent affinity 1–2 orders of magnitude higher than Cu(GHK). Given the high abundance of His residues, the preferred Cu(II) complex of GHK will be ternary Cu(GHK)(N^Im^) rather than binary Cu(GHK). A ternary complex has not been considered before, however, based on our data Cu(GHK) “attached” to bigger peptides or proteins may be the actual form available in blood. Indeed, one may speculate that the isolation of Cu(GHK) from a protein fraction of blood [49] is a consequence of ternary complex formation. A Cu(GHK)(N^Im^) complex may also be the form relevant to other biological effects. For example, Miller et al. found that Cu(GHK) treatment inhibits lipid peroxidation when iron is sourced from ferritin [50]. Cu(GHK) complex was suggested to block physically the ferritin channel disabling the efflux of iron ions. We imply that formation of a ternary Cu(GHK)(N^Im^) complex with one of the numerous His residues present at the channel opening is responsible for this action.

## 3. Materials and Methods

### 3.1. Materials

Gly-His-Lys (#G1887), Imidazole (#I5513), *cis*-UCA, *trans*-UCA, HCl, KNO_3_, HNO_3_, and CuCl_2_ were purchased from Sigma-Aldrich (St. Louis, MO, USA). NaOH was obtained from Chempur (Piekary Slaskie, Poland). ^65^CuO (>99%) was sourced from Cambridge Isotope Laboratories (Tewksbury, MA, USA). The 0.1 M NaOH solution for potentiometric titrations was purchased from POCH (Gliwice, Poland) and standardised via potentiometry using potassium hydrogen phthalate (Merck, Darmstadt, Germany).

### 3.2. UV-Vis & Circular Dichroism Spectroscopy

Spectroscopic measurements were carried out using a J-815 CD spectrometer (JASCO, Easton, MD, USA) and a Lambda 950 UV/vis/NIR spectrophotometer (PerkinElmer, Waltham, MA, USA) in the spectral ranges 270–800 and 270–850 nm, respectively. All experiments were performed at 25 °C in quartz cuvettes with a 1 cm path length.

#### 3.2.1. pH-Metric Titrations

A series of titrations was recorded using UV-vis and CD spectroscopies. For understanding the Cu/GHK interaction we titrated (i) 0.95 mM GHK, 0.8 mM CuCl_2_ and (ii) 10.63 mM GHK, 0.5 mM CuCl_2_, ensuring an excess of GHK over copper ions. Additionally, 4.0 mM *cis*-/*trans*-UCA, 0.8 mM CuCl_2_ was titrated with NaOH in pH ranges 2.05–9.19 and 2.01–5.89 for *cis*- and *trans*-UCA, respectively. At higher pH values, we observed precipitation of copper hydroxide. The final three sets of titrations aimed to understand ternary interaction between (i) 0.95 mM GHK, 0.8 mM CuCl_2_ and 50 mM imidazole and (ii) 0.95 mM GHK, 0.8 mM CuCl_2_, and 6 mM *cis*-/*trans*-UCA. All experiments were performed in the pH-range 2.5–11.5 (unless otherwise stated) with the pH adjusted by addition of minute amounts of NaOH.

#### 3.2.2. Ligand Titrations

Ternary interactions were studied at two pH values, 6.5 and 7.4 by titrating Cu(GHK) with external ligands. A pH of 6.5 mimics the prevailing conditions in the upper layers of the skin, whereas pH 7.4 mimics the blood environment. Titration of imidazole into Cu(GHK) was done using 0.63 mM peptide, 0.5 mM CuCl_2_, and up to 40 molar equivalents of imidazole. This titration was solely performed at pH 7.4 as a control experiment. Additionally, a titration of Cu(GHK) (0.95 mM peptide, 0.8 mM CuCl_2_) with *cis*-/*trans*-UCA was carried out up to 20 molar equivalents of UCA.

### 3.3. EPR Spectroscopy

Samples were prepared at a Cu(II) concentration of 0.5 mM in water. A concentrated stock of ^65^CuCl_2_ was made by dissolving ^65^CuO in 36% *w/w* HCl, followed by removal of excess HCl under heat and addition of milliQ grade water (Millipore, Burlington, MA, USA). A separate sample was prepared for each point in the titration, with the pH measured using a microprobe (Metrohm, Herisau, Switzerland) and adjusted as required using concentrated NaOH. X-band (9.857 GHz) continuous-wave EPR spectra were obtained at room temperature (22 °C) using a Bruker Elexsys E500 spectrometer fitted with a Bruker super-high-Q probehead (ER 4122SHQE, Billerica, MA, USA) and a quartz flat cell (Wilmad, WG-808-Q) for sample containment. Equilibrium was allowed to be established, as ascertained by time-independence of the spectra. The following instrumental settings were used throughout: microwave power, 20 mW; magnetic field modulation amplitude, 5 G; field modulation frequency, 100 kHz; receiver time constant, 40.96 ms; receiver gain, 80 dB; sweep rate, 10 gauss s^−1^; averages, 15. Baseline correction was performed by weighted subtraction of the spectrum obtained using a water blank. Spectral simulations were carried out as previously described [51].

EPR studies of Cu(II) complexes are frequently carried out at low temperatures because the spectra of frozen solutions are independent of the molecular weight. We initially characterised our ternary systems using this approach but found that the binding constant derived from analysis of the frozen solution spectra was almost two orders of magnitude higher than that obtained from the UV-vis and CD spectra. In contrast, room temperature EPR experiments presented herein yielded an excellent quantitative agreement with other spectroscopy. This apparent freezing artefact will be investigated and presented elsewhere.

### 3.4. Potentiometry

Potentiometric measurements were carried out using a 907 Titrando automatic titrator (Metrohm, Herisau, Switzerland), with a Biotrode combined glass electrode (Metrohm, Herisau, Switzerland). The electrode was calibrated daily via titration of 4 mM HNO_3_/96 mM KNO_3_ solution. A 100 mM NaOH solution (carbon dioxide free) was used as a titrant. The 1.5 mL samples were prepared in 4 mM HNO_3_/96 mM KNO_3_ solution. All titrations were performed under argon atmosphere at 25 °C, in the pH range from ca. 2.7 to 11.5. Three to five titrations were used for calculations of protonation constants and Cu(II) formation constants. The obtained data were processed using SUPERQUAD and HYPERQUAD 2008 programs [35,52].

#### 3.4.1. GHK Preparation

Acetate counter-ions present in the commercial GHK preparations can hinder the analysis of potentiometric data. Prior to experiments, we therefore exchanged the acetate counter-ions with TFA via two cycles of peptide dilution in ca. 0.1% TFA, followed by freeze-drying.

#### 3.4.2. Ligands

To calculate protonation constants and the concentration of the stock solution, we prepared samples with ca. (i) 1 mM GHK, (ii) 1–4 mM Im, and (iii) 1–4 mM *cis*-UCA. For each ligand, 3–4 samples were prepared in HNO_3_/KNO_3_.

#### 3.4.3. Binary Complexes

Binary copper complex formation was studied for GHK and *cis*-UCA, respectively, using different molar ratios of CuCl_2_ and ligands. For Cu/GHK, six samples were run in molar ratios of peptide to copper of 1:0.9 (*n* = 2), 1:0.5, 1:0.3 (*n* = 2) and 1:0.2, whereas for Cu/*cis*-UCA four molar ratios were studied: 3:0.8, 4:0.8, 5:0.8, and 8:0.8.

#### 3.4.4. Ternary Complexes

The studied molar ratios of GHK/Cu/Im were: 1:0.9:4, 1:0.9:6 and 1:0.9:8. For each molar ratio two samples were prepared. For GHK/Cu/*cis*-UCA five samples were prepared in ratios of 1:0.9:4; 1:0.9:5; 1:0.9:6; 1:0.9:7; and 1:0.9:8.

### 3.5. Isothermal Titration Calorimetry

Calorimetric titrations were carried out on the Nano ITC Standard Volume calorimeter (TA Instruments, New Castle, DE, USA). The sample cell (950 µL) and syringe (250 µL) were filled with degassed buffered solutions (20 mM HEPES, 100 mM NaCl, pH 7.4) and titrations were performed by ten injections of volume 24 µL added at 1000 s intervals while stirring at 200 rpm, at 25 °C. Solution of 1:0.97 of GHK:CuCl_2_ was titrated with Im or GHK in a range of 0.4 to 40 or 15 molar equivalents, respectively. The concentration of Im in the syringe was 12 mM. The concentrations of GHK in the syringe were 6 or 12 mM. The initial concentrations of GHK in the cell were 0.1, 0.25, 0.35, 0.5, and 1.0 mM, for Im titrations, and 0.25–0.5 mM, for GHK titrations. The obtained data were analyzed with the NanoAnalyze v. 3.11.0 software.

### 3.6. Binding Constant Calculations

*Cu(GHK)_2_*. A self-consistent approach was used to isolate the EPR, UV-vis, and CD spectra of Cu(GHK) and Cu(GHK)_2_ and determine ^c^*K*_7.4_ for the equilibrium CuL + L ⇌ CuL_2_, where L = GHK: (1) An initial guess was made for the value of ^c^*K*_7.4_, which was used to calculate the theoretical speciation of CuL and CuL_2_ at the minimum and maximum value of *n* in the titration Cu/GHK 1:*n* (*n*_min_ ≤ *n* ≤ *n*_max_); (2) The above speciation provided weighting factors that were used to algebraically subtract the spectrum of Cu/L 1:*n*_max_ from that of Cu/L 1:*n*_min_, and vice versa, thus isolating the spectra of CuL and CuL_2_, respectively; (3) Linear combinations of these basis spectra were used to perform a least squares fit of the experimental spectra obtained at all intermediate stoichiometries *n*; (4) The obtained values of [CuL] and [CuL_2_] were used to derive a value of ^c^*K*_7.4_ from the gradient of a plot of [CuL_2_] versus [CuL] × ([Cu]_T_ − [CuL] − 2[CuL_2_]), where [Cu]_T_ is the total concentration of all forms of Cu(II). This experimental value of ^c^*K*_7.4_ was then used as a new guess for ^c^*K*_7.4_, and steps 1–4 were repeated iteratively until the experimental value of ^c^*K*_7.4_ differed from the most recent guess by less than 1%. The above procedure was carried out separately for UV-vis, CD, and EPR data sets. These were then combined into a single plot of [CuL_2_] versus [CuL] × ([Cu]_T_ − [CuL] − 2[CuL_2_]) and a global value of ^c^*K*_7.4_ was determined using least squares nonlinear regression (without outlier removal) implemented in GraphPad Prism version 8.1.1 for Windows (GraphPad Software, San Diego, CA, USA, www.graphpad.com), and the error in ^c^*K*_7.4_ was calculated from the 95% confidence interval derived from the regression analysis.

#### Cu(GHK)(Im), Cu(GHK)(trans-UCA), and Cu(GHK)(cis-UCA)

Using the known formation constants for Cu(GHK) and Cu(GHK)_2_, and the previously isolated spectra of Cu(GHK) and Cu(GHK)_2_, a self-consistent approach was used to isolate the spectrum of Cu(GHK)(Im), Cu(GHK)(*trans*-UCA), and Cu(*cis*-UCA) and determine the value of ^c^*K*_7.4_ and/or ^c^*K*_6.5_ for the equilibrium CuL + A ⇌ CuLA, where L = GHK and A = Im, *trans*-UCA, or *cis*-UCA: (1) An initial guess was made for the value of ^c^*K*_6.5_ or ^c^*K*_7.4_ and the theoretical speciation of CuL, CuL_2_ and CuL was calculated for the minimum and maximum values of *n* in the titration Cu/L/A 1:1.25:*n* (*n*_min_ ≤ *n* ≤ *n*_max_); (2) The above speciation provided weighting factors that were used to algebraically subtract the spectra of CuL and CuL_2_ from the spectrum obtained for Cu/L/A 1:1.25:*n*_max_, thus yielding the spectrum of CuLA; (3) Linear combinations of the CuL, CuL_2_, and CuLA basis spectra were used to perform a least squares fit of the experimental spectra obtained at all intermediate stoichiometries *n*; (4) The obtained values of [CuL], [CuL_2_] and [CuLA] were used to derive a value of ^c^*K*_6.5_ or ^c^*K*_7.4_ from the gradient of a plot of [CuLA] versus [CuL] × ([A]_T_ − [CuLA]), where [A]_T_ is the total concentration of all forms of Im, *trans*-UCA, or *cis*-UCA present in the sample. The experimental value of ^c^*K*_6.5_ or ^c^*K*_7.4_ was then used as a new guess for ^c^*K*_6.5_ or ^c^*K*_7.4_, and steps 1–4 were repeated iteratively until the experimental value of ^c^*K*_6.5_ or ^c^*K*_7.4_ differed from the most recent guess by less than 1%. The binding constants of CuIm*_n_* (*n* = 1–4), Cu(*trans*-UCA)*_n_* and Cu(*cis*-UCA)*_n_* (*n* = 1–2) were too low for these species to influence the results and were therefore not considered. The above procedure was carried out separately for UV-vis, CD, and EPR data sets. These were then combined into a single plot of [CuLA] versus [CuL] × ([A]_T_ − [CuLA]), and a global value of ^c^*K*_6.5_ or ^c^*K*_7.4_ was determined using least squares nonlinear regression (without outlier removal) implemented in GraphPad Prism version 8.1.1 for Windows (GraphPad Software, San Diego, CA, USA, www.graphpad.com), and the error in ^c^*K*_6.5_ or ^c^*K*_7.4_ was calculated from the 95% confidence interval derived from the regression analysis.

## 4. Conclusions

Cu(II)-GHK is a tissue hormone serving as a wound healing factor. Urocanic acid, present as *cis* and *trans* isomers, is the abundant low molecular component of natural moisturizing factor (NMF) of the skin. We have demonstrated that both urocanic acid isomers, and in particular the stable *cis* isomer, form ternary complexes with Cu(GHK). The thorough quantitation of parent and ternary complexes provided the basis for numerical simulations of distribution of Cu(II) ions in NMF. These simulations indicated that Cu(GHK)(*cis*-UCA), but not the binary Cu(GHK) complex is a significant component of the copper pool in NMF. This result, and in general the formation of Cu(GHK)(Im), indicates an urgent need for investigations of biological properties of this and other ternary complexes of GHK.

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
