# Peer review of "Ternary Cu(II) Complex with GHK Peptide and Cis-Urocanic Acid as a Potential Physiologically Functional Copper Chelate"

_ijms, 2020, doi:10.3390/ijms21176190_

Round 1
Reviewer 1 Report
In order to correctly evaluate this work, especially the part of the speciation of GHK and UCA with Cu2+, I need the requirements that I write below. It is very important show all the information about the work in the manuscript (Avoiding the search of previous articles to find relevant information to understand the manuscript) in order to facilitate the work of the referees and also to give all the chemical information for the readers.
- Please, indicate a which formula correspond the species named Cu(II)aq, CuH(GHK), Cu(GHK), CuH-1(GHK), CuH-2(GHK), and CuH2(GHK)2 Cu(cis-UCA)2, Ni(cis-UCA)2 Cu(GHK)(Im) and in general for all the species that are proposing in the manuscript. With this formula the reader doesn’t know through what atoms are coordinated the GHK and the species that they are proposing, in which positions are protonated or deprotonated.
- If the reader want to know to which species are describing for the Cu-GHK the authors must to read the paper that the authors reefer (Biochimica et Biophysica Acta 1526 (2001) 199-210), and in my opinion the present manuscript have to show all the relevant information for the right understanding of it.
Same for table 1 line 146. H2L, HL CuL and CuL2
If the authors want to use that nomenclature they must defined exactly of which species are talking about or, in the best case, a figure with the structures and the corresponding names.
I strongly recommend to the authors draw a scheme in which they show the different speciation with the pH variation. It will be more visual and help to understand the discussion.
- Page 2. Line 76 What “Xaa-His peptides” and “3+1N coordination” mean?. The authors have to define this and other expressions like this in all the manuscript.
- Have the authors confirm the formation of the proposed species in the work by other techniques like HR-MS spectroscopy? With the information obtained by HR-MS the authors avoid speculation about some aspects of the speciation.
Author Response
In order to correctly evaluate this work, especially the part of the speciation of GHK and UCA with Cu2+, I need the requirements that I write below. It is very important show all the information about the work in the manuscript (Avoiding the search of previous articles to find relevant information to understand the manuscript) in order to facilitate the work of the referees and also to give all the chemical information for the readers.
- Please, indicate a which formula correspond the species named Cu(II)aq, CuH(GHK), Cu(GHK), CuH-1(GHK), CuH-2(GHK), and CuH2(GHK)2 Cu(cis-UCA)2, Ni(cis-UCA)2 Cu(GHK)(Im) and in general for all the species that are proposing in the manuscript. With this formula the reader doesn’t know through what atoms are coordinated the GHK and the species that they are proposing, in which positions are protonated or deprotonated.
The information required by Reviewer 1 can be found in the original submission (Table S1, Supplementary Materials). This table contains protonation constants for GHK and complex formation constants for Cu(II)-GHK interaction, pK values of the processes, and protonation events, with detailed information about which functional groups protonate to give the appropriate species. The following statements in the main text draw the reader’s attention to this information: “Potentiometry of Cu/GHK identified six Cu(II) species, namely Cu(II)aq, CuH(GHK), Cu(GHK), CuH-1(GHK), CuH-2(GHK), and CuH2(GHK)2. The calculated logβ values and the ascribed protonation events are presented in Table S1” (lines 63-65). In our opinion, such a description is complete and sufficient. However, to provide further clarification, we have inserted an additional column describing the protonation events in Table 1 (analogously to Table S1) for cis-urocanic acid and its Cu(II) complexes.
- If the reader want to know to which species are describing for the Cu-GHK the authors must to read the paper that the authors reefer (Biochimica et Biophysica Acta 1526 (2001) 199-210), and in my opinion the present manuscript have to show all the relevant information for the right understanding of it.
In our opinion, our response to the previous comment also addresses this comment.
Same for table 1 line 146. H2L, HL CuL and CuL2
If the authors want to use that nomenclature they must defined exactly of which species are talking about or, in the best case, a figure with the structures and the corresponding names.
As abovementioned, we have added to Table 1 a column with the description of protonation events for cis-urocanic acid and its Cu(II) complexes, and clarified that L refers to cis-UCA.
I strongly recommend to the authors draw a scheme in which they show the different speciation with the pH variation. It will be more visual and help to understand the discussion.
The recommended schemes showing the pH-dependent speciation are contained in the Figure 2 (main text) and Figures S1, S13, and S14 (Supplementary Materials) of the original submission.
- Page 2. Line 76 What “Xaa-His peptides” and “3+1N coordination” mean?. The authors have to define this and other expressions like this in all the manuscript.
We have added the following description (line 76): “(...) Xaa-His peptides (where Xaa is any amino acid residue except Pro) (...)”. We have also clarified the meaning of “3+1N coordination” (line 77): “(3+1N coordination, where three nitrogen atoms, NH3+ (N-term), Namide, and NIm, are from the first peptide molecule, and one nitrogen atom is NIm from the second peptide molecule)”.
Have the authors confirm the formation of the proposed species in the work by other techniques like HR-MS spectroscopy? With the information obtained by HR-MS the authors avoid speculation about some aspects of the speciation.
We used in our research potentiometry, EPR, UV-vis, and CD spectroscopies, and calorimetric measurements. Thus we confirmed the speciation by many methods to avoid speculation about the speciation. We think the additional measurements by HR-MS would not improve the manuscript because the measured dissociation constants are not strong enough for such measurement. We are aware of some studies using MS data to confirm the metal ion speciation, but we are also aware of that one has to be very careful with such analysis because MS detection occurs in vacuo, making it difficult to say anything about pH value. Furthermore, because the transition into gas phase is associated with the partial loss of entropic contribution to the complex stability, the Cu(II) complexes often undergo gas phase dissociation or change of stoichiometry. This issue was briefly discussed recently [D. Płonka, W. Bal, The N-terminus of hepcidin is a strong and potentially biologically relevant Cu(II) chelator. Inorg. Chim. Acta 472, 76-81, 2018]. In our opinion, HR-MS measurements would therefore increase speculation.
Reviewer 2 Report
The manuscript by Frączyk and co-workers describes the complex formation processes between the Cu(II)/GHK complex and urocanic acid. The authors reported a thorough characterization of complex equilibria and the results were confirmed by several spectroscopic methods. Nowadays, such in-depth characterization is rare and the results are valuable in the field of biocoordination chemistry. I think that the results are of interest for the readers of the International Journal of Molecular Sciences and so I recommend it for publication.
Some minor issues are listed below:
- The authors reported the formation of CuH-1L (and CuH-2L) species in the Cu(II)/GHK system. In this complex, the binding of OH is expected. However, it was also reported in the literature that the copper(II) and palladium(II) complexes of peptides containing XaaHis motif are able to promote the deprotonation of N1 of imidazole moiety. Could the author provide any proof for the deprotonation of coordinated water molecule?
- For the Cu(II)/GHK/UCA ternary system, the copper(II) is accommodated by the 4 nitrogens in the equatorial plane. However, the carboxylate function of the UCA may contribute to the metal binding in the axial positions. EPR parameters of this complex have not been reported, but it can help to assign the donor groups in the axial positions.
Author Response
The manuscript by Frączyk and co-workers describes the complex formation processes between the Cu(II)/GHK complex and urocanic acid. The authors reported a thorough characterization of complex equilibria and the results were confirmed by several spectroscopic methods. Nowadays, such in-depth characterization is rare and the results are valuable in the field of biocoordination chemistry. I think that the results are of interest for the readers of the International Journal of Molecular Sciences and so I recommend it for publication.
Some minor issues are listed below:
- The authors reported the formation of CuH-1L (and CuH-2L) species in the Cu(II)/GHK system. In this complex, the binding of OH is expected. However, it was also reported in the literature that the copper(II) and palladium(II) complexes of peptides containing XaaHis motif are able to promote the deprotonation of N1 of imidazole moiety. Could the author provide any proof for the deprotonation of coordinated water molecule?
To address this comment, we have added the word “probably” in the statement on line 81: “The first probably corresponds to a water molecule (pK = 9.61) within the equatorial plane of the Cu(GHK) species.” We have also added the following statement (beginning at line 83): “The aforementioned assignment of the deprotonation to a water molecule instead of an imidazole N1 from another Cu(GHK) complex (with concomitant polynuclear species formation) is rationalized by the lack of drastic changes of CD parameters that would be expected upon formation of imidazole-bridged dimeric or tetrameric species [21–23]”, adding the citations of Daniele et al. (J. Chem. Soc. Dalt. Trans. 1991, 2711–2715) and Farkas et al. (J. Chem. Soc., Dalt. Trans. 1984, 3, 611–614).
Because we have added two new references, all reference numbering starting from line 87 has been updated.
- For the Cu(II)/GHK/UCA ternary system, the copper(II) is accommodated by the 4 nitrogens in the equatorial plane. However, the carboxylate function of the UCA may contribute to the metal binding in the axial positions. EPR parameters of this complex have not been reported, but it can help to assign the donor groups in the axial positions.
We agree that carboxylate function of the UCA may contribute to the metal binding in the axial positions, however EPR parameters of the ternary system will not provide insight into axial ligands because their metal-ligand coupling is too small to be resolved using CW-EPR.
Reviewer 3 Report
This is a nice work concerning complexation of copper(II) in the presence of GHK. The authors made a great number of studies to conclude that complex [GHK][Cu(II)][cis-urocanic acid] can form in human tissues which is responsible for known biological effects of GHK and urocanic acid.
The authors provide enormous experimental work: they used UV/vis, CD and EPR spectroscopies to study the interaction between GHK and copper ions in the presence of additional ligands, including protonation processes. Titration calorimetry and potentiometry were used. A number of reactions were studied and binding constants were calculated. No doubt that further investigations of biological properties of ternary GHK complexes should be performed.
This work is interesting and shoud be published in the Journal as it is.
Author Response
The Reviewer did not request any changes.